# Does Deoxynivalenol Affect Amoxicillin and Doxycycline Absorption in the Gastrointestinal Tract? Ex Vivo Study on Swine Jejunum Mucosa Explants

**DOI:** 10.3390/toxins14110743

**Published:** 2022-10-29

**Authors:** Marta Mendel, Wojciech Karlik, Urszula Latek, Magdalena Chłopecka, Ewelina Nowacka-Kozak, Katarzyna Pietruszka, Piotr Jedziniak

**Affiliations:** 1Institute of Veterinary Medicine, Warsaw University of Life Sciences, Nowoursynowka St. 166, 02-786 Warsaw, Poland; 2Department of Pharmacology and Toxicology, National Veterinary Research Institute, Partyzantów 57, 24-100 Pulawy, Poland

**Keywords:** deoxynivalenol, amoxicillin, doxycycline, Ussing chamber, swine jejunum mucosa explants

## Abstract

The presence of deoxynivalenol (DON) in feed may increase intestinal barrier permeability. Disturbance of the intestinal barrier integrity may affect the absorption of antibiotics used in animals. Since the bioavailability of orally administered antibiotics significantly affects their efficacy and safety, it was decided to evaluate how DON influences the absorption of the most commonly used antibiotics in pigs, i.e., amoxicillin (AMX) and doxycycline (DOX). The studies were conducted using jejunal explants from adult pigs. Explants were incubated in Ussing chambers, in which a buffer containing DON (30 µg/mL), AMX (50 µg/mL), DOX (30 µg/mL), a combination of AMX + DON, or a combination of DOX + DON was used. Changes in transepithelial electrical resistance (TEER), the flux of transcellular and intracellular transport markers, and the flux of antibiotics across explants were measured. DON increased the permeability of small intestine explants, expressed by a reduction in TEER and an intensification of transcellular marker transport. DON did not affect AMX transport, but it accelerated DOX transport by approximately five times. The results suggest that DON inhibits the efflux transport of DOX to the intestinal lumen, and thus significantly changes its absorption from the gastrointestinal tract.

## 1. Introduction

Foodstuffs and feed contamination, including simultaneous contamination of agricultural products with numerous mycotoxins and modified mycotoxins, is a frequent and widely recognised worldwide problem [1,2,3,4]. These unavoidable toxins are secondary metabolites produced by different genera of filamentous fungi. They occur on dietary staple foods and fodder, especially cereals, along the whole production chain, including under pre- and post-harvest conditions. In Europe, the most frequently reported mycotoxins and secondary metabolites in feed include deoxynivalenol (DON), zearalenone, ochratoxin A, fumonisin B1, fumonisin B2, and T2/HT2 toxin [3]. Considering pigs’ diet, cereals, including maize and cereal-based products, are probably the most commonly used constituents in feed, supplying most of the animal’s nutrients. Nevertheless, there are mycotoxins in maize called trichothecenes, most importantly zearalenone and DON [2,3,5,6].

Deoxynivalenol is a type B trichothecene produced by Fusarium species. It is believed to be one of the least acutely toxic trichothecenes, but it is highly incident and relevant in animal husbandry [4]. Chronic exposure to low doses of this mycotoxin heavily suppresses the immune response and intestinal functions, induces anorexia, reduces weight gain, and causes neuroendocrine changes [7,8,9,10]. There is sufficient evidence revealing the impairing effect of DON on gut barrier permeability and integrity. The mycotoxin induces the activity of mitogen-activated protein kinases (MAPKs) and decreases the expression of tight junction proteins [11].

Consequently, bacteria and antigens translocation from the lumen of the gut might be intensified [11,12]. Despite the knowledge of DON’s potency to change intestine permeability, little interest has been paid so far to its possible effects on the absorption rate of other xenobiotics at the time of combined exposure [13,14,15,16]. In addition to nutrients, the spectrum of chemicals which might be found in the lumen of the gut due to conscious administration of feed and environmental contaminations include veterinary medicinal products (VMPs), feed additives, fertilisers, plant protection products, air pollutants, and others.

In the case of VMPs, a group of special considerations are antimicrobials. Their use in modern pig production remains one of the elements in maintaining animal health. However, under some conditions, the hazards related to their use could negate their benefits due to the potential risks, including exposure to antimicrobial residues in food or the environment [17,18]. Using antimicrobials might provoke antimicrobial resistance in animal- and human-related bacteria, and thus, compromise animal and human health [19].

Amoxycillin (AMX) and doxycycline (DOX) represent two commonly used antimicrobials for oral application in pigs. Their recommended doses guarantee effectiveness against pathogens and safety of use. Dosing antibiotics (as with all drugs) is based on pharmacokinetic parameters, of which oral bioavailability is one of the key parameters. In the case of orally administered antibiotics in food-producing animals, the level of absorption of the medicine from the gastrointestinal tract affects not only its antibacterial efficacy but also is essential for the safety of food consumers and the environment. In the event of a disturbance in the functioning of the intestinal barrier, the bioavailability of an orally administered antibiotic may change, which in turn may affect the effectiveness and safety of its action. To the best of our knowledge, there is hardly any evidence of the interaction of mycotoxins with antimicrobials within the gastrointestinal tract. An in-depth literature search revealed only one study by Goossens et al. [13] on DON–DOX interaction at the stage of absorption in pigs. Therefore, this study aimed to verify the impact of DON on two antibiotics’ (AMX and DOX) absorption in the intestine isolated from clinically healthy pigs.

## 2. Results

### 2.1. The Effect of Deoxynivalenol on the Viability, Integrity, and Permeability of Jejunum Mucosa Explants

The application of DON at the concentration of 30 µg/mL to the luminal compartment of the Ussing chamber, and incubation of mucosa explants in its presence for 90 min resulted in a significant drop of the transepithelial electrical resistance (TEER) value. It reached only 52.4 ± 0.7 Ohm·cm^2^ at the end of exposure, whereas the control incubation with no mycotoxin resulted in a TEER measurement of 77.1 ± 1.2 Ohm·cm^2^ (Figure 1).

DON caused a remarkable increase in paracellular permeability measured indirectly by the penetration rate of paracellular transport markers. Both Lucifer Yellow (LY) and mannitol (MAN), administered at concentrations of 100 µg/mL, underwent more intense transportation across mucosa explants in intestine specimens treated with DON than in the control chambers (Figure 2 and Figure 3).

The flux of LY and MAN amounted to 89.5 ± 3.2 and 306.0 ± 8.6 ng/min/cm^2^, respectively, in the presence of DON, and to 38.3 ± 1.7 and 217.3 ± 6.5 ng/min/cm^2^, respectively, in the absence of the mycotoxin. The flux of the transcellular transport marker (caffeine—CAF) did not change when mucosa explants were incubated in a DON-containing buffer. The addition of mycotoxin caused CAF penetration through intestine explants at the level of 2.9 ± 0.2 µg/min/cm^2^, whereas in the control trial, the flux came to 2.6 ± 0.1 µg/min/cm^2^ (Figure 4).

Moreover, the use of DON did not provoke any cytotoxicity measured by LDH leakage. The activity of LDH detected in the buffer amounted to 4.9 ± 0.2% and 4.7 ± 0.2% of total LDH activity in the presence and absence of the mycotoxin, respectively (Figure 5).

### 2.2. The Effect of Amoxicillin and Doxycycline on the Viability, Integrity, and Permeability of Jejunum Mucosa Explants

The single exposure of mucosa explants to either AMX (50 µg/mL) or DOX (30 µg/mL) did not provoke a significant change in TEER values during 90 min of incubation. The final measurement of TEER indicated 76.8 ± 3.1 and 70.1 ± 2.2 Ohm·cm^2^ for AMX- and DOX-treated jejunum tissues, respectively. In contrast, no addition of antibiotics caused a TEER reading of 77.1 ± 1.2 Ohm·cm^2^ (Figure 1). The use of AMX did not provoke any significant change in the penetration of paracellular transport markers because the flux of LY and MAN amounted to 41.8 ± 5.3 and 230.6 ± 5.0 ng/min/cm^2^, respectively (Figure 2 and Figure 3). Similarly, AMX did not affect the penetration rate of the transcellular transport marker. The flux of CAF was measured as 2.4 ± 0.1 µg/min/cm^2^ in the presence of this antibiotic and 2.6 ± 0.1 µg/min/cm^2^ when the explants were incubated in AMX-free medium (Figure 4). Likewise, the addition of DOX did not modify the intensity of CAF penetration across mucosa explants. DOX revealed the tendency to increase the intensity of transportation of paracellular transport markers. The flux of LY and MAN reached 58.7 ± 5.15 and 247.8 ± 18.9 ng/min/cm^2^, respectively (Figure 2 and Figure 3). Additionally, none of the tested antibiotics increased the release of LDH compared to the control conditions. The enzyme activity in the KRB amounted to 5.2 ± 0.2 and 5.4 ± 0.2% of total LDH activity for AMX and DOX, respectively (Figure 5).

### 2.3. The Effect of Combined Exposure to Deoxynivalenol and Amoxicillin or Doxycycline on the Viability, Integrity, and Permeability of Jejunum Mucosa Explants

The combined exposure of mucosa explants to DON and one of the antibiotics did not provoke a more profound alteration in intestine integrity and permeability than the mycotoxin used solely. Simultaneous exposure to AMX + DON or DOX + DON did not alter the magnitude of the TEER drop compared to the effect of DON alone. TEER readings were at the same level and amounted to 54.2 ± 2.9, 54.9 ± 1.5, and 52.4 ± 0.7 Ohm·cm^2^ for AMX + DON, DOX + DON, and DON, respectively (Figure 1). In the case of the penetration of paracellular and transcellular transport markers through mucosa explants, there were no remarkable differences between tissue samples incubated only in the presence of DON and those incubated in a cocktail of DON and one of the antibiotics. The flux of LY came to 87.7 ± 10.6, 101.9 ± 7.8, and 89.5 ± 13.2 ng/min/cm^2^ for AMX + DON, DOX + DON, and DON, respectively (Figure 2). The penetration of MAN ranked at 317.3 ± 15.7, 318.4 ± 12.6, and 306.0 ± 8.6 ng/min/cm^2^, respectively, for AMX + DON, DOX + DON, and DON-containing KRB, respectively (Figure 3). Similarly, the extra addition of AMX or DOX did not cause any significant change in CAF penetration across jejunum mucosa in comparison to the effect of DON (Figure 4). However, the rate of CAF penetration was significantly higher in the presence of DOX + DON when compared to the control trial. The cytotoxicity measured in the LDH leakage test was at the same level for explants incubated in DON-containing incubation medium with and without antibiotics (Figure 5).

### 2.4. The Effect of Deoxynivalenol on Amoxicillin and Doxycycline Penetration across Swine Jejunum Explants

The penetration rate of AMX across jejunum mucosa explants amounted to 18.8 ± 2.5 ng/min/cm^2^. The intensity of AMX transportation did not change in the presence of DON because antibiotic flux remained very similar, i.e., at the level of 16.6 ± 1.2 ng/min/cm^2^ (Figure 6A). In the case of DOX, the basic penetration rate (in the absence of the toxin) was 0.7 ± 0.1 ng/min/cm^2^. The combined exposure to DOX and DON caused a 5-fold increase in the antibiotic penetration rate, which finally came to 3.8 ± 0.5 ng/min/cm^2^ (Figure 6B).

## 3. Discussion

Due to their ubiquitous presence, mycotoxins affect the health of humans and animals consuming plant-based food and feeds. Financial losses caused by mycotoxins occur because of decreased crop yields, loss of crop value, effects on domestic animal productivity, and human health impacts. In the framework of the presented study, the toxic effect of DON on pig jejunum was confirmed. The results obtained with the alternative model of jejunum mucosa explants isolated from routinely slaughtered, clinically healthy adult pigs delivered evidence of DON potency to decrease mucosa barrier integrity and increase its permeability. A significant drop of TEER values over the time of tissue incubation in the presence of the mycotoxin proved progressively declining integrity of intestine explants (Figure 1), which confirms previous observations from cell- and tissue-based experiments [11,20,21]. Modification of TEER values indicate disturbances in epithelial barrier function or the transcellular permeability of ions [11]. Since the rate of caffeine, a transcellular transport marker, translocation remained unaffected in the presence of DON (Figure 4), it is concluded that the toxin does not affect this transportation pathway. Our finding of increased paracellular permeability measured by enhanced penetration of LY and MAN, two markers of paracellular transport, from luminal to contraluminal compartment (Figure 2 and Figure 3) is in line with remarks of others [11,20,22]. A significant difference between the results generated herein and data collected by others is the relatively high dose of DON engaged by us. However, the differences in sample collection, especially the use of adult pigs as explant donors, seem to justify the discrepancies, as discussed previously [23]. Nevertheless, presented data confirm the potency of DON to increase intestine permeability and affect the absorption rate of chemicals and other antigens present in the lumen of the gut at the same time as the mycotoxin.

Most of the toxicological data refers to the effects of chemical contaminants when present alone; however, animals are usually exposed to cocktails of numerous compounds, which might impact their health [14]. In animal production, concurrent oral exposure to mycotoxins and veterinary medicinal products cannot be ruled out. The One Health strategy turns attention to the consequences of the combined presence of antimicrobials and intestine-affecting mycotoxins, in particular DON. To understand the interactions between DON and antimicrobials, we have selected two popular antibiotics used to control infectious diseases in pigs, i.e., amoxicillin and doxycycline. First-line antibiotics are among the most commonly used antibiotics in food-producing animals, including pigs.

Moreover, these antibiotics are often used orally after mixing with feed or dissolved in drinking water. According to the new AMEG categorisation, both amoxicillin and doxycycline belong to Category D “Prudence”, meaning the risk to public health associated with the use in veterinary medicine of substances included in this category is considered low [24]. To maintain the usefulness of AMX and DOX, it is crucial to keep their dosing adequate for effectiveness and, simultaneously, to cause no risk of remaining residues in animal-origin products. For both studies, antibiotics have a potency of augmented absorption under favourable conditions like those induced by gut barrier permeability enhancers, including DON. Enhanced absorption of antibiotics from the gastrointestinal tract might influence their pharmacokinetic parameters. Consequently, their pharmacodynamic activity might pose the risk of prolonged presence of antibiotics in animal bodies, contributing to the development of bacterial resistance, environmental persistence, and ecotoxicity.

According to the results presented herein, none of the tested antibiotics possesses the ability to influence intestine integrity and permeability, and they also do not contract the disturbances induced by DON (Figure 1, Figure 2, Figure 3, Figure 4 and Figure 5). Intestine disturbances provoked by DON did not affect the intensity of AMX penetration across mucosa explants under proposed experimental conditions (Figure 6A), but increased DOX transport by about five times (Figure 6B).

To the best of our knowledge, there is no other trial analysing AMX or other aminopenicillins’ representative absorption intensity in the presence of mycotoxins in pigs. Regarding DOX, Goossens et al. [13] observed that the plasma concentration of DOX was remarkably higher in the pigs that received DON-contaminated feed supplemented with the mycotoxin binder. In pigs exposed solely to DON, there was only a small, statistically insignificant, increase of mean plasma DOX concentration compared to control animals [13]. The possible explanation for the discrepancies between quoted data and our results is using different experimental conditions and doses of the mycotoxin in both studies. In our study, we observe a clear effect of DON on the transport of DOX under conditions controlled for the presence of all substances that may affect the absorption process. Goossens points out in his research, that feed ingredients other than DON can influence the absorption of DOX [13].

Based on the obtained results, it is impossible to define the mechanism of the observed interactions between DON and DOX. The analysis of markers flux in the presence of both DON and DOX, indicating that the intensification of paracellular transport is more probable than the enhancement of transcellular absorption. However, it is possible to hypothesise that DOX and DON compete against access to glycoprotein P (GPP).

Studies in the field of pharmacokinetics indicate that DOX, applied in therapeutic doses, absorbs from swine gut according to the first-order kinetics [25,26,27,28]. This means that the intensity of DOX absorption depends only on the concentration of the antibiotic, and there is no need to consider other mechanisms involved in the transportation process. Even if an additional transport mechanism for DOX is recognised (efflux involving GPP) and happens in pigs during the absorption phase in the gut, its impact on total DOX transportation is not limited by the accessibility of transport mechanism (the presence of such mechanism does not need to be included in the kinetic equation for DOX absorption).

The observations and conclusions from pharmacokinetic studies align with our experiment’s conditions because the concentrations of DOX and AMX in the mucosal chamber are similar to those measured in vivo in the intestine. The concentrations of the antibiotics applied in the study presented herein, i.e., AMX = 50 µg/mL, DOX = 30 µg/mL, represent the preliminary concentrations of those drugs in the gut lumen when administered orally with drinking water in pigs. The recommended dose of AMX in pigs is 10–20 mg/kg b.w. every 12 h [29]. Moreover, the recommended dose of DOX amounts to 10 mg/kg b.w. every 12 h [30]. Assuming treated pigs’ average daily water intake comes to 0.1 L/kg b.w., AMX and DOX should be applied at the concentration of 150 µg and 100 µL per 1 mL of drinking water, respectively. When an antibiotic is drunk once by a pig, it gets diluted 2–4 fold by the content of the stomach and intestine before it gets absorbed. Hence, the expected concentration of selected antibiotics at the beginning of the jejunum absorption phase amounts to 50 and 30 µg/mL for AMX and DOX, respectively.

Our observation of increased intensity of DOX transportation across mucosa explant in the presence of DON suggests the involvement of a mechanism which amplifies the penetration of DOX from mucosal to serosal chamber. These findings cannot be explained only as a consequence of increased mucosa barrier permeability (induced by DON) and subsequent enhancement of the paracellular transport of DOX because the first-order absorption kinetic of DOX (observed in pharmacokinetic studies) means that only the concentration of DOX determines absorption. In other words, the assumptions of first-order kinetics result in unlimited DOX penetration by paracellular transport, which already occurs under control conditions (DON-free medium). Another possibility includes the switching out of a transport mechanism, which is in opposition to the absorption of DOX and which, in the presence of DON, occurs as an important component determining the penetration of DOX across the mucosa barrier. Therefore, it is speculated that in the presence of DON, the mechanism of efflux transport of DOX is revealed as a significant factor influencing the intensity of DOX absorption from the gut. The increase of DOX transportation in the presence of DON possibly depends on transport by GPP. There is evidence that GPP transports DON, and acting on GPP may affect the transport of other drugs [31,32,33]. Martinez et al. (2013) observed that concurrent exposure of IPEC-J2 cells to fosfomycin (580 µg/mL) and DON (1 µg/mL) resulted in a remarkably higher intracellular concentration of the antibiotic in the enterocytes, confirming the potency of DON to enhance drug penetration from the lumen of the gut [16]. DOX is also transported via GPP [34,35]. GPP is responsible for the transport of DOX from the enterocytes’ cytoplasm to the gastrointestinal tract’s lumen (efflux transport).

The results presented herein demonstrate that there is a possible competition between DON and DOX against GPP. The mycotoxin, as a compound of higher affinity to GPP, might block the efflux of DOX if the efflux mechanism of DOX is switched out. The transportation of the antibiotic increases what is indicated by the enhanced flux of DOX across the intestine. Regardless of the mechanism of DON’s impact on DOX absorption, and bearing in mind the assumptions of pharmacokinetic model describing DOX absorption from the gut, the first-order kinetic must be ruled out at the co-occurrence of DOX and DON.

## 4. Conclusions

In summary, DON intensifies the transportation of DOX across the porcine gut wall but displays no impact on AMX absorption. The increase of DOX penetration might result from reduced availability of the GGP efflux transport system for the antibiotic. Such an effect of DON can cause remarkable changes in DOX pharmacokinetics and affect the pharmacodynamic properties and safety of DOX in pigs.

The results presented herein justify further in vivo research on DON’s impact on DOX absorption, bioavailability, and excretion to realise consumer exposure, environmental persistence, and ecotoxicity.

Bearing in mind that climate change globally, the fungal population and mycotoxin patterns in different regions and crops are changing [36]. Their impact on animal health and the potency of inducing interaction with other xenobiotics, including antimicrobials, should not be underestimated and requires more in-depth investigation.

## 5. Materials and Methods

### 5.1. Chemicals

Caffeine (CAF), Cytotoxicity Detection Kit (LDH) Roche, D-mannitol (MAN), D-Mannitol Colorimetric Assay Kit, disodium fumarate, deoxynivalenol (DON), L-glutamate, lucifer yellow (LY), and sodium pyruvate were obtained from Sigma Aldrich (St. Louis, MO, USA). All inorganic salts required for preparing Krebs Bicarbonate Buffer (KRB), ethanol, and glucose were purchased from Avantor (Gliwice, Poland). Amoxicillin trihydrate (AMX) and doxycycline hyclate (DOX) were generously donated by the pharmaceutical company Biofaktor Sp. z o.o. (Skierniewice, Poland) The quality of antibiotics was consistent with the monographs of the European Pharmacopoeia and corresponded to the quality of active substances used in the production of veterinary drugs.

Tissue transportation, preparation, and incubation were performed in Krebs Bicarbonate Buffer (KRB) containing 108 mM NaCl, 4.7 mM KCl, 1.8 mM Na_2_HPO_4_, 0.4 mM KH_2_PO_4_, 15 mM NaHCO_3_, 1.2 mM MgSO_4_, 1.25 mM CaCl_2_, 11.5 mM glucose, 4.9 mM L-glutamate, 5.4 mM disodium furmate, and 4.9 mM sodium pyruvate at pH 7.4, and saturated with oxygen using a 95%/5% O_2_/CO_2_ mixture by gassing for 60 min [37].

For HPLC and LC-MS/MS analysis, acetonitrile, methanol, and formic acid (HPLC grade) were obtained from Avantor Chemicals (Radnor, PA, USA), trichloracetic acid and heptafluorobutyric acid were obtained from Sigma Aldrich (St. Louis, MO, USA).

### 5.2. Tissue Preparation

Healthy female and male (60%:40%) adult landrace and large white pigs of approx. 100 kg body weight subjected to routine slaughtering were used for the collection of intestinal tissue. In total 18 animals were used as tissue donors. Segments of the jejunum (approx. 150 cm aboral to pylorus) were obtained and handled as described in other research [21,22,23,38,39,40],. Briefly, jejunum pieces of approx. 50 cm in length were gently incised immediately after stunning and flushed to remove intestine content. Next, the samples were immersed in ice-cold KRB and brought to the laboratory where they were subjected to the preparation. Firstly, they were cut into pieces of 10–20 cm and opened longitudinally. Secondly, the serosa and muscular layers were carefully stripped from the mucosa using forceps. Eventually, four mucosa explants were gained from each animal. Each resulting sheet of mucosa with attached submucosa was mounted separately between two Ussing-type half chambers (1.54 cm^2^ tissue exposure area). Jejunum sheets were bathed on luminal (mucosal) and contraluminal (serosal) surfaces in 10 mL of KRB, maintained at pH 7.4 and 37 °C. Mucosa explants were continuously oxygenated on luminal and contraluminal surfaces with a 95%/5% O_2_/CO_2_ mixture delivered by gas lift. The complete system was firstly preincubated for 10 min for the equilibration of the tissue. Afterwards, the incubation medium was replaced by fresh KRB in the serosal chamber, and KRB supplemented with LY in concentration 100 µg/mL, mannitol 100 µg/mL, and caffeine 100 µg/mL (KRB + LY + MAN + CAF) in the mucosal chamber.

### 5.3. Measurement of the Viability, Integrity, and Permeability of Mucosa Explants

The viability of the swine jejunal mucosa sheet was analysed by measuring several markers directly after preincubation (time 0), and 30, 60 and 90 min afterward. The integrity of mucosa explants was controlled by measuring transepithelial electrical resistance (TEER) using a Millicell ERS-2 Epithelial Volt-Ohm Meter (Merck GA, Darmstadt, Germany). The prerequisite of TEER readings greater than 70 Ω·cm^2^ at time 0 was settled to verify jejunum preparations’ usefulness for other parts of the experiment. Additionally, the integrity and viability of the explants were verified by measuring the flux of LY, MAN, and CAF over time from the luminal to the contraluminal compartment. To assess the possible tissue damage caused by the presence of active proteases or experiment duration, the leakage of lactate dehydrogenase (LDH) to both the mucosal and serosal compartments was recorded.

### 5.4. Ex Vivo Exposure of Swine Jejunum to Deoxynivalenol and Antibiotics

Four explants of jejunum mucosa were prepared from each pig and fixed separately in Ussing-type chambers. All explants underwent 10-min preincubation followed by 90-min incubation in KRB (serosal chamber). In the mucosal chamber, pure KRB was replaced by: (i) KRB + LY + MAN + CAF with neither addition of DON nor AMX nor DOX (control condition), (ii) KRB + LY + MAN + CAF supplemented with DON (30 µg/mL), (iii) KRB + LY + MAN + CAF containing DON (30 µg/mL) and AMX (50 µg/mL) or DOX (30 µg/mL), (iv) KRB + LY + MAN + CAF containing AMX or DOX (50 and 30 µg/mL, respectively). After the KRB exchange in the luminal compartment, the incubation was continued for another 90 min. TEER measurement and sample collection (800 µL) for later LY, MAN, CAF, LDH, DON, and AMX or DOX assays were carried out at times 0, 30, 60 and 90 min after the onset of incubation.

### 5.5. Analyses

LY was analysed directly in samples using an FLx800 Microplate fluorescence reader (BioTek Instruments, Inc., Winooski, VT, USA) at excitation wavelength 485 nm and emission wavelength 530 nm. According to the manufacturer’s instructions, MAN concentration and LDH activity were determined using a D-Mannitol Colorimetric Assay Kit and Cytotoxicity Detection Kit (LDH) Roche.

CAF concentration was analysed by the HPLC-UV method as follows. The sample was centrifuged, and the supernatant was filtered through a 0.22 µm filter. Next, 20 μL of filtered samples were injected into the HP 1100 HPLC system consisting of a quaternary pump, thermostatic autosampler, sample thermostat, column thermostat, and diode array detector. The system was fitted with a Nucleosil^®^ 120-5C18 HPLC (250 mm × 4.6 mm, 5 µm) column (Supelco, Bellefonte, PA, USA). The mobile phase was methanol. The flow rate was 1 mL/min. Detection was performed at a wavelength of 254 nm (ref. 360 nm). The analytes were identified with retention times of pure reference standards. The reference standards were also used to prepare a standard solution to establish calibration curves. Chromatographic peak areas of the analytes were measured using the integrator software for the HPLC system (Agilent ChemStation Rev. A.06.01 [403] Hewlett Packard Company, Palo Alto, CA, USA).

For the extraction of DOX and AMX, 100 µL of cell fluid collected from the serosal chamber was placed into a 1.5 mL microcentrifuge tube, then 20 µL of IS and 100 µL of 5% trichloracetic acid (DOX)/0.1% formic acid (AMX) were added, mixed, diluted, and centrifuged at 14,500× *g* for 5 min. The supernatant was filtered through a 0.22 µm PVDF syringe filter into an LC vial for UHPLC-MS/MS analysis.

The analysis of antibiotics was performed using ultra-high-performance liquid chromatography with detection by triple quadrupole mass spectrometry (UHPLC-MS/MS: Shimadzu Nexera X2, Kyoto, Japan coupled with QTRAP 4500, AB Sciex, Framingham, MA, USA.). The following parameters were used: temperature—450 °C; curtain gas (N_2_)—20; nebuliser gas (N_2_)—60; collision gas (N_2_)—medium; auxiliary gas—65; ion spray voltage—4500 V (Table 1).

The chromatographic separation assay was performed using an Agilent InfinityLab Poroshell 120 EC-C18 (150 mm × 2.1 mm, 2.7 µm) column (Agilent, St. Clara, CA, USA) with an octadecyl guard column (2 × 4 mm) maintained at 35 °C. The mobile phase consisted of 0.025% heptafluorobutyric acid (A) and acetonitrile (B) at a flow rate of 0.3 mL/min, with an injection volume of 5 µL. Gradient elution for AMX was conducted as follows: 0–1 min 95% A, 5–8 min 20% A, 8–8.01 min 50% A, and finally from 8.01 to 10 min back to 95% A; for DOX 0–5 min 50% A, 5–6 min 10% A, 6–8 min 10% A, and finally from 8.01 to 10 min back to 95% A The total run time in both cases was 10 min.

The method has been validated. LOQ values (DOX 0.01 ng/L, AMX 0.02 ng/L), linearity, reproducibility (CV: DOX 5.6–8.7%, AMX 7.7–10.0%), and recovery (DOX 85–105%, AMX 90–97%) were determined.

The results of LY, MAN, CAF, AMX, and DOX penetration across intestine mucosa explants are expressed as mass flux. The amount of LDH leakage into the incubation media is expressed as a percentage of total LDH, which was analysed after explants homogenisation in ice-cold KRB with a Potter S Homogenizer (B. Braun Biotech International, Berlin, Germany) for 2 min at 1000 rpm.

### 5.6. Statistical Analysis

The experimental result is expressed as means ± SEM. Differences between groups were statistically determined using one-way ANOVA followed by Tukey’s multiple comparisons test or t-test if only two groups were compared. Results were considered statistically significant when *p* < 0.05.

Analyses were performed using GraphPad Prism version 8.0.0 for Windows, GraphPad Software, San Diego, CA, USA, www.graphpad.com.

## Figures and Tables

**Figure 1 toxins-14-00743-f001:**
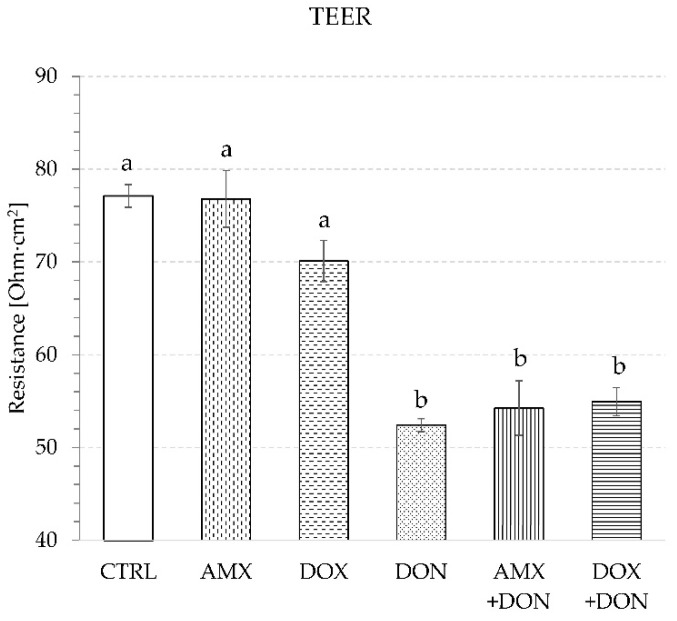
TEER of intestine explants measured after 90 min incubation in buffer supplemented with: amoxicillin—AMX, doxycycline—DOX, deoxynivalenol—DON, and combination AMX + DON or DOX + DON, or CTRL—control condition without antibiotics and DON. Bars show the mean of the 6 replicates ± SEM (standard errors of the mean). Different letters above the bars indicate a statistically significant difference at *p*-value < 0.05.

**Figure 2 toxins-14-00743-f002:**
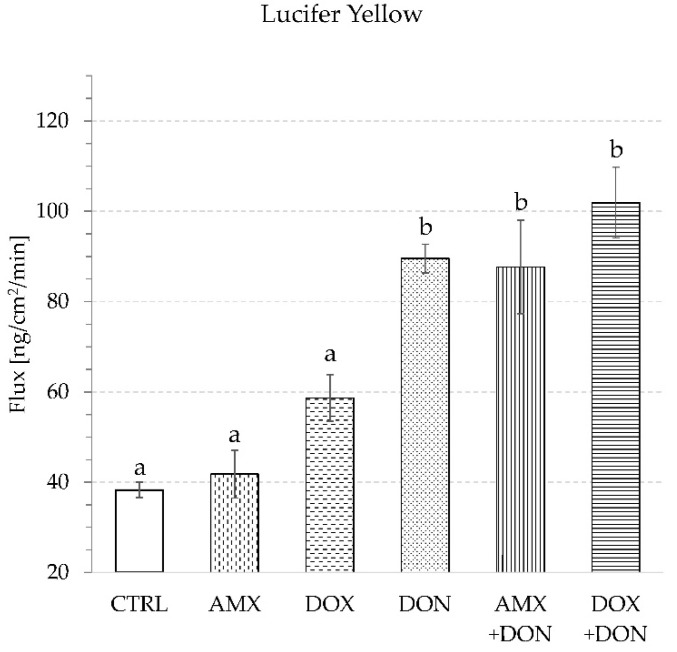
Lucifer Yellow transport through intestine explants during 90 min of incubation in buffer supplemented with: amoxicillin—AMX, doxycycline—DOX, deoxynivalenol—DON, combination AMX + DON or DOX + DON, or CTRL—control condition without antibiotics and DON. Bars show the mean of the 6 replicates ± SEM (standard errors of the mean). Different letters above the bars indicate a statistically significant difference at *p*-value < 0.05.

**Figure 3 toxins-14-00743-f003:**
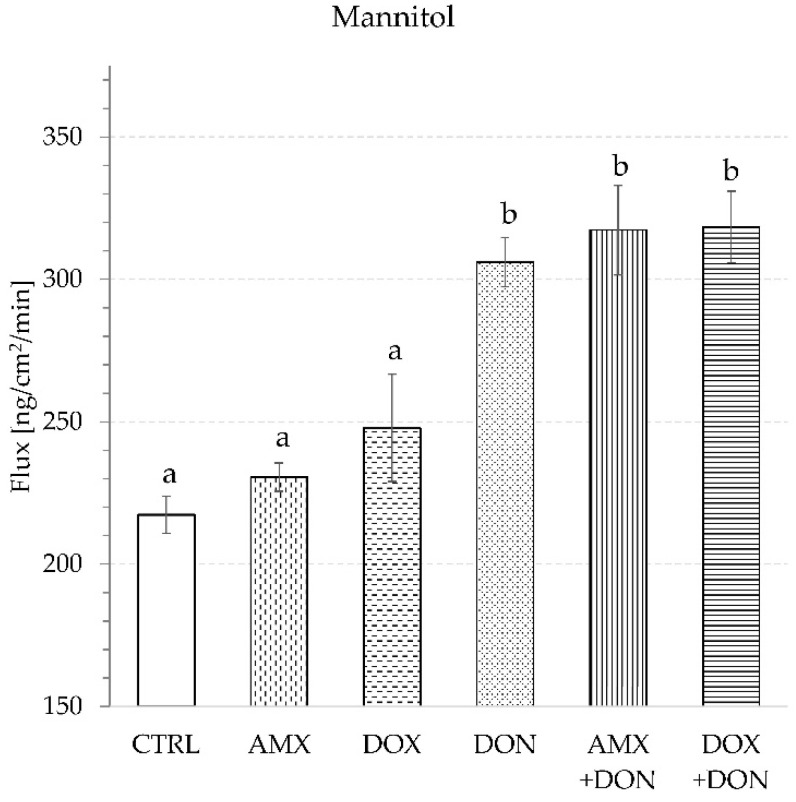
Mannitol transport through intestine explants during 90 minincubation in buffer supplemented with: amoxicillin—AMX, doxycycline—DOX, deoxynivalenol—DON, combination AMX + DON or DOX + DON, or CTRL—control condition without antibiotics and DON. Bars show the mean of the 6 replicates ± SEM (standard errors of the mean). Different letters above the bars indicate a statistically significant difference at *p*-value < 0.05.

**Figure 4 toxins-14-00743-f004:**
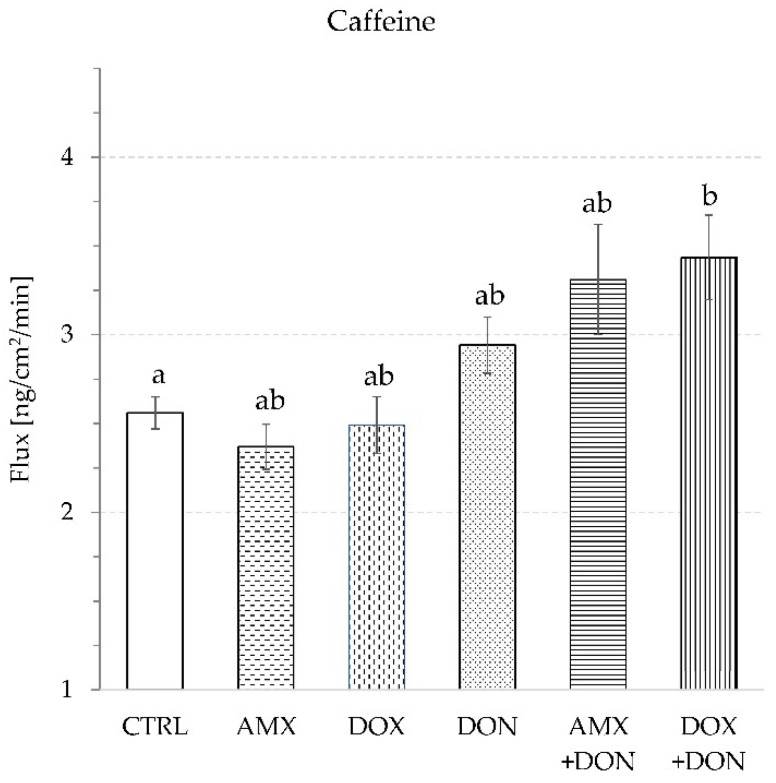
Caffeine transport through intestine explants during 90 min incubation in buffer supplemented with: amoxicillin—AMX, doxycycline—DOX, deoxynivalenol—DON, combination AMX + DON or DOX + DON, or CTRL—control condition without antibiotics and DON. Bars show the mean of the 6 replicates ± SEM (standard errors of the mean). Different letters above the bars indicate a statistically significant difference at *p*-value < 0.05.

**Figure 5 toxins-14-00743-f005:**
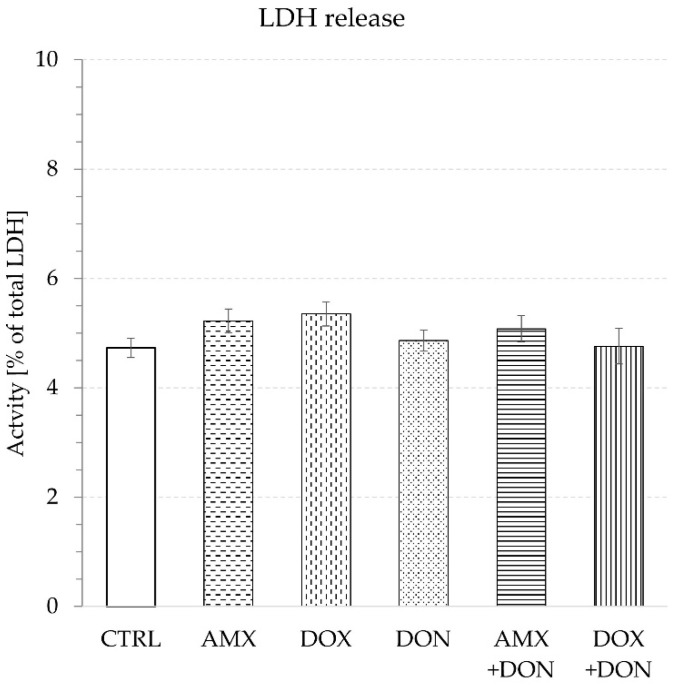
Relative LDH activity in the luminal compartment of intestine explants at 90 min of incubation in buffer supplemented with: amoxicillin—AMX, doxycycline—DOX, deoxynivalenol—DON, combination AMX + DON or DOX + DON, or CTRL—control condition without antibiotics and DON. The LDH activity measured after explant homogenisation was taken to be 100%. Bars show the mean of the 6 replicates ± SEM (standard errors of the mean).

**Figure 6 toxins-14-00743-f006:**
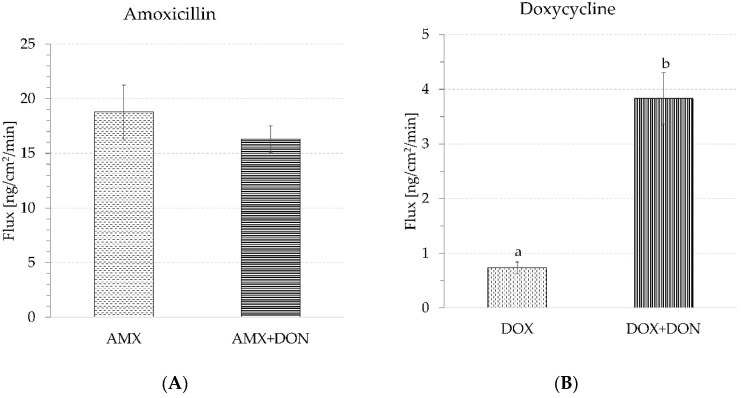
Effect of DON on the transport of antibiotics through intestine explants during 90 min of incubation. (**A**) Transport of AMX used alone (AMX) and in the presence of DON (AMX + DON); (**B**) Transport of DOX used alone (DOX) and in the presence of DON (DOX + DON). Bars show the mean of the 6 replicates ± SEM (standard errors of the mean). Different letters above the bars indicate a statistically significant difference at *p*-value < 0.05.

**Table 1 toxins-14-00743-t001:** Detailed MS/MS conditions of doxycycline and amoxicillin analysis.

Analyte	Parent Ion M + H^+^ [*m/z*]	Daughter Ions [*m/z*]	DP [V]	CE [eV]	Dwell Time [ms]
Doxycycline (444.4 g/mol)	445.4	428.0; 154.0	60	24; 41	250
Amoxicillin (365.4 g/mol)	366.1	349.0; 208.0	45	12; 18	250

## Data Availability

Not applicable.

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
