# Peer review of "Does Deoxynivalenol Affect Amoxicillin and Doxycycline Absorption in the Gastrointestinal Tract? Ex Vivo Study on Swine Jejunum Mucosa Explants"

_toxins, 2022, doi:10.3390/toxins14110743_

Round 1

Reviewer 1 Report

Review on

“Does deoxynivalenol affect amoxicillin and doxycycline absorption in the gastrointestinal tract? Ex vivo study on swine jejunum mucosa explants?”

toxins 103390

Summary

The authors used porcine jejunal explants in an Ussing chamber. Antibiotics (DOX and AMX) and DON were applied on the luminal side and fluxes of markers (Lucifer yellow, mannitol, caffeine) were measured in the basal compartment of the chamber. High DON loads triggered an increased flux of low molecular weight components (LY, mannitol) whereas caffeine transport was largely insensitive versus DOX, AMX or DON. DON enhanced the flux of DOX, but not the AMX flux.

Line 81: 30 µg/mL is a very high dosage of DON and will surely not reached in vivo. The authors justified this dosage by previous work of Mendel et al 2019, line200-204. Please explain this justification a bit more in detail at his point in the discussion.

Line 92: Are the data normal distributed?

Line112: But CAF flux was increased with DON +DOX?

Line195: I think, this is an important finding. Despite the high DON load the transport mechanism was not hampered.

Line233: The term “transport” is justified later in the discussion (Line272-302).

Line297: Did the authors thought of an experimental setup in which DOX was applied on the serosal side of the explant and DON on the luminal?

Line 406: “For extraction of DOX and AMX, 100µl of cell fluid..”. Did the authors mean the KRB from the basal side of the Ussing chamber?

Minor:

Line 15: TEER

Line 48: Please add relevant literature.

Line 84: “Om”? Please us “Ohm” if the Greek letter is not available.

Line 340: Numbers of cites are missing

Line 344: REM: Body temperature of pig is 39°C.

Author Response

Dear Reviewer,

Thank you very much for your time and careful reading of the manuscript. We greatly appreciate your comments and believe that the revision of the manuscript according to your suggestions will enhance its quality and make it acceptable for the publication.

All corrections introduced in the manuscript are highlighted in red.

Please find our replies to your comments:

Summary

The authors used porcine jejunal explants in an Ussing chamber. Antibiotics (DOX and AMX) and DON were applied on the luminal side and fluxes of markers (Lucifer yellow, mannitol, caffeine) were measured in the basal compartment of the chamber. High DON loads triggered an increased flux of low molecular weight components (LY, mannitol) whereas caffeine transport was largely insensitive versus DOX, AMX or DON. DON enhanced the flux of DOX, but not the AMX flux.

Line 81: 30 µg/mL is a very high dosage of DON and will surely not reached in vivo. The authors justified this dosage by previous work of Mendel et al 2019, line200-204. Please explain this justification a bit more in detail at his point in the discussion.

Short additional information has been added. As describes and compared in details previously, the model used for our study is quite unique due to the fact of using adult pigs (of approx. 100 kg b.w.) as explant donors. The difference in effective doses of DON might possibly be explained by a tolerance or compensation of adult pigs to DON or compensatory mechanisms induced by previous negative effect of DON to re-establish intestinal homeostasis. Based on the results of monitoring studies on feed contamination by mycotoxins in Europe, it must be assumed that adult pigs had been chronically or continuously exposed to DON before they were slaughtered, whereas piglets used as donors of mucosa explants by others had ingested feed containing controlled doses of DON and only for a  relatively short time, as they were slaughtered latest at the age of 3 months.

Line 92: Are the data normal distributed?

Yes, all results are normally distributed. The distribution is checked with the Kolmogorov-Smirnov test. Additionally, before the ANOVA test is performed, the statistical program runs the Brown-Forsythe test and the Bartlett test. Both tests confirmed that the variances in all compared groups were equal.

Line112: But CAF flux was increased with DON +DOX?

Thanks for the observation. Additional comment was added in subchapter 2.3, lines 167-168.

Line195: I think, this is an important finding. Despite the high DON load the transport mechanism was not hampered.

The importance of the observation was emphasized additionally in the discussion, lines: 250-252.

Line233: The term “transport” is justified later in the discussion (Line272-302).

The authors decided to use the terms: transport and penetration interchangeably. In contrast, we were very careful when using the term “absorption” as it is much more complex process in vivo than simple penetrations/transport.

Line297: Did the authors thought of an experimental setup in which DOX was applied on the serosal side of the explant and DON on the luminal?

In the experimental setup of the project, we only considered concurrent exposure to DON and antibiotics. However, we agree that it would be interesting to analyze how DOX (or other antibiotic) kinetics changes in a situation when DON had been absorbed earlier and affect cells (enterocytes) performance.

Line 406: “For extraction of DOX and AMX, 100µl of cell fluid..”. Did the authors mean the KRB from the basal side of the Ussing chamber?

The information has been particularized.

Minor:

Line 15: TEER - corrected

Line 48: Please add relevant literature. - added

Line 84: “Om”? Please us “Ohm” if the Greek letter is not available. - corrected

Line 340: Numbers of cites are missing – some more were added

Line 344: REM: Body temperature of pig is 39°C. The incubator used to prepare all the media as well as the water-coat of Ussing Chambers were set at 39°C. It was just typing mistake in the manuscript. It has been corrected.

Reviewer 2 Report

I consider the submitted publication to be of high quality and suitable for publication in your magazine. I only have a few minor comments.

Keywords are repeated in the title

The authors write: Healthy adult landrace and large white pigs of approx. 100 kg body weight subjected to routine slaughtering was used for the collection of intestinal tissue.

Please indicate what was the composition of the pigs in terms of sex.

From how many pigs in total were intestinal samples taken

The authors write: Segments of the jejunum (approx. 150 cm aboral to pylorus) were obtained and handled as described before (Mendel et al., 2019; Sjöberg et al., 2013, Westerhout et al., 2014).

Please describe this procedure very briefly in a few sentences.

Author Response

Dear Reviewer,

First of all, we would like to express our gratitude for reading and suggesting improvements to the manuscript. All of the points you raised were carefully analysed and addressed. Below please find detailed replies to your comments.

All corrections introduced in the manuscript are highlighted in red.

We hope that in the present form the manuscript fulfills your expectations and can be accepted for the publication.

I consider the submitted publication to be of high quality and suitable for publication in

your magazine. I only have a few minor comments.

Keywords are repeated in the title. The authors used the same words on purpose to turn the attention of the audience to main topics tackled in the manuscript.

The authors write: Healthy adult landrace and large white pigs of approx. 100 kg body

weight subjected to routine slaughtering was used for the collection of intestinal tissue.

Please indicate what was the composition of the pigs in terms of sex.

There was quite even representation of both sexes. Details are added in the manuscript.

From how many pigs in total were intestinal samples taken Additional information was added.

The authors write: Segments of the jejunum (approx. 150 cm aboral to pylorus) were

obtained and handled as described before (Mendel et al., 2019; Sjöberg et al., 2013,

Westerhout et al., 2014). Please describe this procedure very briefly in a few sentences.

Short description was added.

Reviewer 3 Report

The paper deals with deoxynivalenol (DON) whose presence in feed may modify intestinal barrier's permeability. The studies were conducted using jejunal explants from adult pigs. The topic is interesting since the modification of the intestinal barrier integrity may affect the absorption of antibiotics commonly used in animals and so far, there is only little evidence about this aspect. To this regard, changes in transepithelial electrical resistance, the flux of transcellular and intracellular transport markers and the flux of antibiotics across of explants were measured.

On the basis of the results achieved, the authors demonstrated that DON is capable to inhibit the efflux transport of DOX to the intestinal lumen significantly changing its absorption from the gastrointestinal tract.

Prior to potential publication, I invite the authors to clarify the following aspects:

-The language is readable although it may be somewhat improved.

-Lines 73-74: The authors must critically report the works carried out so far proving evidence of the interaction of mycotoxins with antimicrobials within the gastrointestinal tract.

Lines 393-394. The UHPLC-MS/MS instrument used for the analysis of antibiotics must be described in detail.

Lines 396: N2 “2”  must be subscript.

Lines 406-410 must be places before UHPLC-MS conditions. Further, did the authors employ a validated protocol?

The Table reported in section “5.5. Analyses” is not mentioned. The authors may consider to delete it and report the content in the text. In addition, the exact mass for both analytes must be reported.

Author Response

Dear Reviewer,

First of all, we would like to express our gratitude for reading and suggesting improvements to the manuscript. All of the points you raised were carefully analysed and addressed. Below please find detailed replies to your comments.

All corrections introduced in the manuscript are highlighted in red.

We hope that in the present form the manuscript fulfills your expectations and can be accepted for the publication.

The paper deals with deoxynivalenol (DON) whose presence in feed may modify intestinal barrier's permeability. The studies were conducted using jejunal explants from adult pigs. The topic is interesting since the modification of the intestinal barrier integrity may affect the absorption of antibiotics commonly used in animals and so far, there is only little evidence about this aspect. To this regard, changes in transepithelial electrical resistance, the flux of transcellular and intracellular transport markers and the flux of antibiotics across of explants were measured.

On the basis of the results achieved, the authors demonstrated that DON is capable to inhibit the efflux transport of DOX to the intestinal lumen significantly changing its absorption from the gastrointestinal tract.

Prior to potential publication, I invite the authors to clarify the following aspects:

-The language is readable although it may be somewhat improved. When revising the manuscript some re-wording were introduced.

-Lines 73-74: The authors must critically report the works carried out so far proving evidence of the interaction of mycotoxins with antimicrobials within the gastrointestinal tract. The statement has been supported by adding additional information.

Lines 393-394. The UHPLC-MS/MS instrument used for the analysis of antibiotics must be described in detail.

The UPLC-MS/MS instrument was described.

Lines 396: N2 “2”  must be subscript. Corrected.

Lines 406-410 must be places before UHPLC-MS conditions. Further, did the authors employ a validated protocol?

The information about validation results was added.

The Table reported in section “5.5. Analyses” is not mentioned. The authors may consider to delete it and report the content in the text. In addition, the exact mass for both analytes must be reported.

The table was mentioned and exact masses for antibiotics were added

Round 2

Reviewer 3 Report

The authors have adequately addressed all comments and the improved paper can be now accepted in the present form.